# Differential Entropy: An Appropriate Analysis to Interpret the Shape Complexity of Self-Similar Organic Islands

**DOI:** 10.3390/ma14216529

**Published:** 2021-10-29

**Authors:** Stefano Chiodini, Pablo Stoliar, Pablo F. Garrido, Cristiano Albonetti

**Affiliations:** 1Consiglio Nazionale delle Ricerche—Istituto per lo Studio dei Materiali Nanostrutturati (CNR-ISMN), Via P. Gobetti 101, 40129 Bologna, Italy; Stefano.Chiodini@iit.it; 2Center for Nano Science and Technology, Fondazione Istituto Italiano di Tecnologia, Via G. Pascoli 70, 20133 Milan, Italy; 3National Institute of Advanced Industrial Science and Technology (AIST), Tsukuba 305-8565, Japan; p.stoliar@aist.go.jp; 4Departamento de Fisica de Aplicada, Facultade de Fisica, Universidade de Santiago de Compostela, E-15782 Santiago de Compostela, Spain; Pablo.Fernandez@usc.es

**Keywords:** in situ imaging, atomic force microscopy, differential entropy, fractal dimension, sexithiophene, Ehrlich-Schwoebel barrier

## Abstract

Differential entropy, along with fractal dimension, is herein employed to describe and interpret the shape complexity of self-similar organic islands. The islands are imaged with in situ Atomic Force Microscopy, following, step-by-step, the evolution of their shape while deposition proceeds. The fractal dimension shows a linear correlation with the film thickness, whereas the differential entropy presents an exponential plateau. Plotting differential entropy versus fractal dimension, a linear correlation can be found. This analysis enables one to discern the 6T growth on different surfaces, i.e., native SiO_x_ or 6T layer, and suggests a more comprehensive interpretation of the shape evolution. Changes in fractal dimension reflect rougher variations of the island contour, whereas changes in differential entropy correlates with finer contour details. The computation of differential entropy therefore helps to obtain more physical information on the island shape dependence on the substrate, beyond the standard description obtained with the fractal dimension.

## 1. Introduction

The failure of standard atomistic theories to completely describe Organic Thin Films (OTFs) has already shone light on the richer physics of organic molecular systems. The growth of OTFs, due to their complex shapes and internal degrees of freedom, gives rise to peculiar behaviours, such as rapid roughening [1], roughening transition [2], and thickness dependent lattice unit vectors [3]. The Ehrlich-Schwoebel barrier (ESB) [4,5] has been recognized as the main factor causing a layer-by-layer, or pyramidal, growth of the film [6], but the lack of a standard theory of the molecular film growth makes its evaluation difficult. In the literature, only a few examples of molecular simulations [7,8] and experimental measurements [6,9,10] provide, or attempt to provide, insights on its calculation.

To overcome such limitations, the morphology of OTFs has been deeply investigated, looking for the building blocks of the molecular film growth [11]. Most of the molecules employed in these experiments are small organic molecules deposited on weakly interacting substrates, e.g., SiO_2_ [12]. Such configurations are fundamental in the fabrication of organic electronic devices [13,14] where a strong correlation between morphology and electrical performance was extensively proved [15,16,17,18].

In general, OTFs on SiO_2_ follow the Stranski-Krastanov (SK) growth mode [19,20], a mixed scenario involving both Frank–Van der Merwe (FM, two-dimensional films) and Volmer–Weber growth modes (VW, three-dimensional films) [21]. At the SiO_2_ interface, the OTF is characterized by two or three complete FM layers [21]. These layers, also known as monolayers (MLs), are complete and composed of quasi upright standing molecules [22,23,24,25,26,27]. After that, the growth switches to VW and three-dimensional structures form [19].

In the past, scaling laws were employed to study, statically [1,28,29] and dynamically [19,26,30,31], the morphological evolution of OTFs for increasing thickness, proving the self-affinity of the OTFs surface profile [10,32,33,34,35,36].

On the other hand, a geometrical approach is used when OTFs are ultrathin (≤1 ML), i.e., at the SiO_2_ interface. In this initial stage, the average area, superficial density, surface coverage, and critical size of mounds and/or islands were measured [37,38,39,40,41,42]. Their self-affinity was studied with the fractal dimension [43,44], thus connecting the geometrical shape of mounds and/or islands to their growth evolution in the sub-monolayer regime [45,46,47,48,49,50]. Such a geometrical shape, as originally explained by Ehrlich [4], is connected to the ESB, and thus to the fractal dimension. On this basis, Hlawacek et al. [6] have experimentally measured the ESB for para-sexiphenyl (6P) mounds, suggesting, however, that the inorganic growth theory, applied to organic growth, needs to take into account some additional effects dependent on the type of molecule employed.

To correctly describe the island border complexity, information entropy appears to be the right mathematical concept. This approach has already been used in computer vision where information entropy allows the recognition of two-dimensional shapes (or patterns) through their complexity [51,52,53].

Originally adopted by Shannon [54], the term ‘information entropy’ can generate misunderstandings and confusion. He defines entropy as a measure of the uncertainty of data in an information channel, so the highest value corresponds to “missing information” [55]. As reported in the literature [56,57], this mathematical entropy is not correlated to the thermodynamic one, although the latter one has been successfully applied to both theoretical [58,59,60] and experimental [61] statistical mechanics.

In modern science, information entropy is often associated with the fractal dimension, with both being measurements of complexity. Among others, applications can be found in geology [62], biology [63], and medicine [64]. In the field of OTFs, only in a study of fullerene film fractal dimension and entropy were singly calculated [65].

This work aims to fill this gap by measuring island shape complexity through the differential entropy, i.e., the information entropy extended to continuous variables. Such entropy is measured by evaluating the probability distributions of local angles along the island contours [52]. This analysis increases the classification efficiency of the island shapes beyond fractal dimension. Mathematical relationships linking differential entropy and fractal dimension can help to discern growth phenomena through geometrical parameters [66].

Such measurements are performed with in situ Atomic Force Microscopy (AFM) that allows step-by-step imaging of the growth evolution [19]. Thus, the shape complexity can be measured with a high accuracy of film thickness. To our knowledge, this is the first example reported up to now where the shape complexity is monitored in situ, in UHV, and with time.

## 2. Materials and Methods

### 2.1. Sample Preparation

The growth of 6T OTFs on native SiO_x_ (Boron doped, *p*-type, 5–10 Ω·cm) was performed subliming 6T polycrystalline powder (Sigma-Aldrich, used as received) through an Organic Material Effusion (OME) Knudsen cell (Dr. Eberl MBE-Komponenten GmbH, Weil der Stadt, Germany) in Ultra-High Vacuum (UHV—base pressure 2 × 10^−10^ mbar). Two experiments at different substrate temperatures *T_S_* were performed, viz. 25 and 50 °C. The deposition flux was kept constant to ~1 Å∙min^−1^. The OTF thickness is measured in MLs where a monolayer (ML) is a layer of ordered and packed molecules almost orthogonal to the SiO_x_ surface that completely covers it (*Θ* = 1). Accordingly, each layer composing the OTF can be expressed in terms of an equivalent surface coverage *Θ* [15]. The maximum thickness was fixed to 6.5 ML, as measured at 25 °C, where molecular desorption can be considered negligible [19]. In the case of 6T, each monolayer corresponds to a thickness of 2.4 nm [22], so the maximum OTF thickness is 16.9 nm.

### 2.2. AFM Imaging

The growth of 6T OTFs was followed in situ and step-by-step with a commercial UHV system (Omicron GmbH, Taunusstein, Germany), combining an OME Knudsen cell and an AFM microscope (Omicron GmbH, Taunusstein, Germany, VT-UHV SPM XA 50–500 K). Relatively stiff cantilevers (Nanosensors, resonance frequency of ~300 kHz, and spring constant of ~10 N∙m^−1^) were employed in Non-Contact AFM (NC-AFM). Sequential AFM images were taken every ~0.25 ML, i.e., every four minutes of deposition. The scan area was selected to 6 × 6 μm^2^, the largest possible scan size granting stable NC-AFM images. *T_S_* was set to 25 and 50 °C to provide a statistically relevant number of islands because, as shown in a previous work [19], higher *T_S_* do not provide enough islands. The same work reports also two different populations of islands at early stage of the growth: smaller islands with a high density and bigger islands with a low density [19,67]. The oblique position of the OME cell, together with the non-hermetic closure of its shutter, are responsible for such bimodal growth. In fact, when the inclination angle of the OME cell, with respect to the unit vector normal to the substrate plane, is larger than 40° (in our case 60°), the substrate collects molecules from the lobe-shaped vapour clouds coming from larger emission angles (see Figure 43 in reference [68]), a phenomenon increased by the distribution of the powder along the crucible wall [69]. In this work, bigger islands at each *T_S_* are excluded from the morphological analysis.

### 2.3. Image Processing and Data Analysis

As proposed by Mandelbrot [70], the perimeter–area (P–A) relationship can be applied to determine the fractal dimension *D_f_* of the perimeter-line of the islands, analysing not an individual island but a group of them. In order to do that, fractality should hold in a “wide enough” size-range [71], where two orders of magnitude are usually considered wide enough. Through this method, 6T islands imaged by AFM on a large scan-range (viz 20 × 20 μm^2^) were demonstrated to be a set of self-similar objects [29]. Due to both our limited scan range (6 × 6 μm^2^) and the average island size for each *T_S_*, the images of this work can yield erroneous *D_f_* values if calculated with the standard P–A method because the size-range of fractality is not “wide enough” and the number of islands is limited [72]. To overcome this problem, *D_f_* was calculated with the box counting method [73], for which the size-range of images in pixels, i.e., 1024 × 563 px^2^, is “wide enough” to hold fractality on, at least, three orders of magnitude.

The topographic images were processed with Gwyddion software [74]: (1) data were levelled by the mean plane subtraction; (2) paraboloidal background was removed; (3) rows were aligned by means of the median of differences; (4) horizontal scars were corrected; (5) mean values filter (3 × 3 px^2^) was applied to remove any possible topographic artefact.

Amongst all the images that follow the growth step-by-step (at least four images for each ML), only those below the aggregation regime for each ML were analysed (partial surface coverage *Θ_p_*~0.5 ML + *n* where *n* = 0, 1, 2, and 3 ML) [75]. This choice avoids an erroneous fractal dimension calculus with the lacunarity method [43] and satisfies the condition of well-separated islands. Accordingly, the islands are firstly marked by thresholding, and the selected islands are then filtered by area in order to remove larger islands induced by the bimodal growth (see Figure 1a). The resulting mask is extracted and used to statistically evaluate the average area of the islands, thus determining the largest box size (closeness power of 2) employed in the box counting method (see Figure 1b, see Appendix A). As required for fractal calculus in ImageJ [76], the mask is changed to 8 bit and binarized. Then, the fractal dimension is calculated by using power of 2 box sizes, i.e., 2^0^, 2^1^, 2^2^,…, thus reducing the error in *D_f_* (Figure 1c) [73].

To go a step further, the fractal dimension, which, at least in OTF theories, appears to be only a geometrical parameter, the differential entropy *S* of islands is introduced [77] (see Appendix A):(1)$$S=-\int_{0}^{180}p\left(\alpha \right)ln\left[p\left(\alpha \right)\right]d\alpha$$
where the continuous random variable *α*, with probability density function (pdf) *p*(*α*), ranges from 0 to 180° (degree).

The differential entropy extends the Shannon entropy concept to a continuous probability distribution [78]. All histograms generated and analysed in this study have a consistent number of bins set to 180 (bin size Δα = 1°), hence *S* ranges from 0 to ~5.2 nats (see Appendix A).

In order to calculate *S*, AFM images were firstly segmented by ad hoc image segmentation procedure consisting of three steps (Figure 2b–d, see Appendix A). The original AFM image (Figure 2a) is firstly binarized by using the mask procedure described above, obtaining a two-level bitmap image with a black background and white islands (Figure 2b). The contour of the islands is then traced by using the “trace bitmap” function of Inkscape software [79] (Figure 2c). This function detects island contours, “tracing” them using a series of lines and cubic Bezier curves. Finally, traced contours were turned into polygonal chains composed of small segments with “adapted” lengths (Figure 2d, see Appendix A).

Once islands are segmented, α for each segment is obtained by measuring the angle between the semi straight line joining the centre *C* of the island and the segment centre *P* (Figure 2e, see Appendix A). Lastly, the differential entropy *S* of Equation (1) is calculated as the pdf area of the bar chart, where *p*(*α*) is the ratio between the sum of segment lengths with angle α and the island contour length, normalized with respect to the bin Δ*α* (see Appendix A).

## 3. Results

Figure 3 shows AFM images of 6T OTFs deposited on native silicon oxide [37]. For each *T_S_*, i.e., 25 (left column, a–d) and 50 °C (right column, e–h), an image for each layer is represented at roughly half of the layer growth and up to 4 ML at most. The sequence of AFM images illustrates the step-by-step and in situ evolution of OTFs.

At each *T_S_*, the same 6 × 6 μm^2^ region was scanned enabling to follow the growth of every single 6T island (cp. to Section 2.2). The column on the left shows the 25 °C case (a–d) while the column on the right corresponds to the 50 °C case (e–h). Several observations can be done: (i) regardless *T_S_*, OTFs grew in a SK growth mode; (ii) at 25 °C, 6T islands have an average area smaller than at 50 °C [80], (iii) the island superficial density is clearly affected by the temperature, with a higher density at 25 °C with respect to 50 °C [38,40], and (iv) the island shape at 50 °C appears to be more dendritic with respect to 25 °C. These results agree to the OTF theory [29,81], confirming the proper OTFs preparation by OMBD in UHV.

As described in Section 2.2, the bimodal growth is caused by the oblique position of the OME cell, and its effect is particularly evident in the first ML (Figure 3a,e). Smaller islands with higher superficial density are considered consistent to the 6T aggregates formed by OMBD in UHV, thus larger islands are excluded from the morphological analysis (cp. to Figure 1a).

Amongst several morphological parameters, the fractal dimension *D_f_* is commonly used to quantify the average island shape [29,44,82]. Specifically, *D_f_* is expected to be close to 1 for a perfectly round island [83], while it increases towards 2 for islands with highly ramified contours [84], passing through randomly ramified aggregates grown by the Diffusion Limited Aggregation (DLA) condition (*D_f_* = 1.71) [85]. Taking Figure 4 of reference. [86] as a reference, the island shape is termed compact (*D_f_* → 1), dendritic (1 < *D_f_* < 2) or fractal (*D_f_*~1.71) when its shape is regular (e.g., a circle) [87], has (few or many) branches characterizing preferred growth directions [44], or is highly randomly ramified [82,88], respectively. In situ AFM imaging enables to follow step-by-step the OTF growth [19], hence *D_f_* versus *Θ*.

Due to the limits discussed in Section 2.3, *D_f_* of the 6T islands is calculated by using the box counting method [73]. The fractal dimension increases linearly versus *Θ* within each ML (see dashed lines in Figure 4). Its value shifts from compact islands with *D_f_* ranging from 1.5 to 1.6, like pentacene islands [89], to dendritic ones with *D_f_*~1.75 that is close to 1.71, the mathematical limit characterizing fractal islands formed by DLA (see horizontal dashed-dotted line in Figure 4) [43,90]. This effect is particularly evident for sequential in situ AFM images at 50 °C where the same islands evolve from smooth to dendritic shapes (see Figure 4, right column). The direct visual comparison between AFM images at 25 and 50 °C also explains why *D_f_* is lower at 25 °C than at 50 °C for any *Θ*: lower *T_S_* promoted islands with smoothed shapes.

The slopes of linear fits, below the aggregation regime, decrease for increasing MLs, suggesting an evolution of island shape versus MLs. The first ML is excluded from linear fitting since it is described by only two *Θ* points. This trend can be explained by following the evolution of *D_f_* for comparable partial surface coverage *Θ_p_*, e.g., ~1.3 ML, ~2.3 ML, and ~3.3 ML (see dotted lines in Figure 4). The progressive linear increasing of *D_f_* for increasing *Θ_p_* causes an increase of *D_f_* at the initial *Θ_p_*, i.e., ~1.1 ML, ~2.1 ML, and ~3.1 ML, while *D_f_* is constant to ~1.75 for *Θ_p_* just below the aggregation regime of each ML, i.e., ~1.5 ML, ~2.5 ML, and ~3.5 ML. In reason of that, a progressive slope decreasing for increasing MLs is observed. This trend is independent from *T_S_* because, although *D_f_* at 50 °C is larger than at 25 °C, *D_f_* versus *Θ* slopes decrease linearly with the same slope (see Appendix A).

The differential entropy *S* provides additional information on the geometrical evolution of 6T islands for increasing deposited molecules.

In the information theory, the Shannon entropy *SE* of a discrete random variable is the average level of uncertainty inherent to the variable possible outcomes [54] (see Appendix A). The minimum uncertainty occurs for a known event (probability *p* = 1), so the entropy *SE* is zero bits. Conversely, the maximum uncertainty occurs for equiprobable events with *p* = *n*^−1^ where *n* is the number of bins, hence *SE* is at its maximum. Other values of *p* give different entropies *S* between these two limits.

The differential entropy *S* extends the Shannon entropy concept to a continuous variable, thus the probability histogram turns into a probability distribution function (pdf) that characterizes the amount of information contained in the continuous variable [91]. In our case, the probability density function *p*(*α*) (see Equation (1)) describes the probability to find the angle α within the range [0, 180] degree. Besides the difference between discrete and continuous variables, *SE* measures an absolute (information) entropy, whereas *S* measures a relative entropy, i.e., a change of uncertainty [92]. For instance, Fhionnlaoich et al. [93] have used *SE* to characterize the size distributions of monodisperse nanoparticles. Nevertheless, they classify the nanoparticle population into a set of categories computing (absolute) *SE*. Conversely, angle distributions computed from AFM images in these experiments are treated as a continuous variable, returning a (relative) differential entropy *S*. Specifically, *S* measures the relative change of islands’ entropy as the film growths, but it cannot provide any conclusion on its absolute value.

As described in Section 2.3, *α* pdfs obtained analysing AFM images show characteristic features (see Appendix A): (i) compact islands are characterized by high angular information with a Gaussian distribution and a lower *S* (see Figure 5, pdf on the left); (ii) dendritic islands are characterized by low angular information with a *quasi*-rectangular distribution, i.e., angles α are *quasi*-equiprobable, and a higher *S* (see Figure 5, pdf on the right); (iii) regardless of *Θ* and *T_S_*, pdfs are characterized by a constant baseline (blue dashed line in both pdfs); (iv) pdfs at 25 °C mainly show a single Gaussian and, sometimes, a convolution of two, while pdfs at 50 °C are balanced between single Gaussian and *quasi*-rectangular distributions. This trend suggests that 6T islands are more compact at 25 °C with respect to the ones at 50 °C.

In most cases, pdfs can be fitted with Gaussian (left panel in Figure 5, dashed black fit) and uniform functions (right panel in Figure 5, *y* = *a* where *a* is the constant probability—dashed black line). The differential entropy *S* (in nats) is ½*ln*[(2·π·*e*)·*σ*^2^] and −*a*·*ln*(*a*)·(*x_f_* − *x_i_*) for Gaussian and uniform functions [94], respectively. Specifically, *σ* is the standard deviation of the Gaussian distribution, whereas (*x_f_* − *x_i_*) is defined by integral limits of Equation (1), i.e., 180. In sporadic cases, pdf is the convolution of two Gaussians and *S*~½*ln*[(2·π·*e*)·*σ*^2^] + A, where A is a constant defined in reference [77] (By applying these functions to pdfs of Figure 5, *S* is: (left panel) ½*ln*[(2·π·*e*)·28^2^]~4.75 nats plus an additional entropy due to a constant baseline with *a* = 0.00028 giving −180·0.00028·*ln*(0.00028)~0.41 nats. The total *S* for compact islands is therefore ~5.16 nats. This value is slightly larger (6%) than the calculated one—see Appendix A—because of the differential entropy, used for a continuous variable, approximates the exact solution of the Shannon entropy used for a discrete variable; (right panel) in average *a* = 0.0055 therefore −180·0.0055·*ln*(0.0055)~5.15 nats. Both are within the *S* range 0 and ~*ln*(180)~5.2 nats).

Regardless of the pdf shape, a constant baseline is always present or, in other words, α has always a probability different from 0 to undertake all possible angles in the range [0, 180] degree. Such probability depends on the island shape: small baseline for compact islands and vice versa—high baseline for dendritic islands (see caption of Figure 5). In agreement with *D_f_* measurements, *T_S_* promotes more dendritic islands and larger pdf baselines at 50 °C rather than at 25 °C.

As shown in the Appendix A, *S* depends on the average islands size *A_px_* (in px, i.e., the average number of pixels included in each island) and its value saturates after the critical size *A_px_* = 190 px at this scan-size (1024 × 563 px^2^, 6.0 × 3.3 μm^2^). Additionally, 6T islands at early stages of the growth, i.e., *Θ* = 0.25 ML, have *A_px_* of ~200 px and ~440 px for 25 and 50 °C, respectively. These average areas permit to deem *S* stable.

As shown in Figure 6, the differential entropy *S* depends on both *Θ* and *T_S_*, growing faster in the first ML, and then saturating in the next MLs. Data are empirically fitted with the exponential *plateau* function:(2)$$S=S_{sat}-\Delta S\cdot e^{-\frac{\Theta -\Theta_{T}}{\Theta_{C}}}$$
where *S_sat_* is the saturation value of *S*, Δ*S = S_sat_* − *S*_0_ where *S*_0_ is the initial islands entropy, *Θ_C_* is the thickness constant, and *Θ_T_* is a horizontal offset (plot translation). As explained in the Appendix A, *S* is a scale-dependent parameter and *S*_0_ is the initial islands entropy at a given scale-size, i.e., the differential entropy correspondent to the constant baseline. Strictly speaking, *S* → 0 when island average area tends to 0. Therefore, *S*_0_ cannot be the differential entropy at *Θ =* 0, i.e., when there are not molecules deposited on the surface. As the average area of islands is reduced, their contours tend to be circular and smaller in length, thus *S* decreases. Accordingly, plots of Figure 6 are expected to shift towards *x*-axis due to progressively smaller *S_sat_* and *S*_0_.

The saturation value of *S* is lower than ~5.2 nats, the maximum entropy for this system, for both *T_S_* (see Table 1). As expected, *S* saturates at a higher value for a higher *T_S_* because islands are more dendritic, hence they are less dendritic at 25 °C.

In the exponential *plateau* model, the thickness constant *Θ_C_* (in ML) describes how rapidly island shape, i.e., *S*, saturates for increasing deposited molecules. This process results faster at 50 °C rather than at 25 °C due to the increased diffusivity of molecules on the surface induced by *T_S_*. In this empirical function, *S* reaches the 95% of the final asymptotical value when *Θ* = *3Θ_C_*, i.e., ~0.8 ML at 25 °C, and ~1.1 ML at 50 °C thus, in average, *S* saturates at ~0.9 ML that corresponds to the plot region where fitting curves are overlapped (see Figure 6). The large difference of *S*_0_ for 25 and 50 °C (see Table 1) corresponds to compact or dendritic shapes, respectively (at the scan-size analysed in these experiments).

A graphical extrapolation of *S*_0_ from fitting curves is clearly weak since it is supported by only two datapoint for each *T_S_* (only two thickness *Θ* satisfy the experimental condition of separated islands at *Θ* < 1 ML) and because the fitting function has the highest derivative at *Θ* = 0.

Tentatively, the initial entropy *S*_0_ could be dependent on the pdfs constant baselines. Averaging the constant baselines of pdfs sets (11 pdfs for each *T_S_*), *S*_0_ is (2.8 ± 1.1) nats for 25 °C and (3.2 ± 1.3) nats for 50 °C. These intervals include *S*_0_ graphically extrapolated from fitting curves but, although more statistically relevant, this result is not supported by a theoretical explanation.

Since *S* and *D_f_* are complexity measurements, a general monotonic positive relationship is expected. Figure 7 shows two distinctive regions for growing films fitted linearly: one correspondent to the first ML (green dashed line) and one for the other MLs up to the fourth one (red dashed line). In each region, independently on *T_S_* (e.g., blue and purple data points in the first ML), entropy and fractal dimension are linearly correlated. Linear fits have two markedly different slopes: (i) in the early stage of the growth, data follow a linear trend with slope Δ*S*/Δ*D_f_* ≈ 1, so the entropy change has a one-to-one correspondence to the change of the island shape; (ii) in the next MLs, data collapse around a linear fit line almost horizontal to the *x*-axis with slope Δ*S*/Δ*D_f_* ≈ 0.2 thus, despite to a large variation in the island shape (*D_f_* varies from ~1.35 to ~1.7), the island entropy changes slightly.

As reported in the literature [95,96,97], the angle entropy against the fractal dimension was used to discriminate physical/chemical parameters with different origins but, to date, a theoretical framework for explaining such correlations does not exist.

In 6T OTFs experiments, a tentative interpretation is possible, ascribing different slopes to different surface diffusion of 6T molecules on either native SiO_x_ or 6T MLs (cp. to Discussions section). According to OTF theory, surface diffusion energy depends on the diffusing layer: as reported in reference [98], diffusion of 6T molecules on native SiO_x_ involves a higher diffusion energy than for 6T molecules on a 6T ML (this is true also for next 6T MLs). A functional dependence of *S* versus *D_f_* slope with respect to the diffusion energy is therefore expected and, specifically, the slope should be inversely related to the ease of diffusion of the molecules on the underneath layer.

## 4. Discussions

In general, the island shape is a result of the competition between two kinetic processes: edge diffusion and monomer incorporation [49]. Edge diffusion is characterized by the time taken by a monomer to diffuse along the island edge for finding an energetically favourable binding site. Monomer incorporation is characterized by the average time interval between two consecutive monomer incorporation events. The effect of frequent monomer incorporation is opposite to that of edge diffusion.

Islands formed in the DLA regime are taken as a reference system for the island growth. The monomer incorporation in the DLA regime is faster than the edge diffusion, leading to growth instabilities and fractal island shapes [44].

Both the incoming monomer flux, i.e., the deposition rate, and the surface area accessible to monomers via surface diffusion [99] determine the island growth rate at an island edge site. Monomer incorporation is enhanced by high flux, while the surface diffusion is characterized by a diffusion length determined by Voronoi tessellation [38]. Dendritic and fractal islands are formed when the diffusion length of monomers is small and islands are largely separated whereas, at the opposite (viz large diffusion length and small separation), compact or dendritic islands are formed. In the DLA growth, where fractal islands form, long protruding branches are associated with largely separated islands and a small diffusion length.

Based on the possibilities described in the aforementioned scenario, the island shape is modulated by both substrate and deposition rate. Specifically, largely separated islands are obtained on hydrophobic substrates that interact less with monomers, enabling their diffusion on the substrate surface (large diffusion length) [100]. The H passivated Si(111), H-Si(111), is an example of *quasi* hydrophobic substrate employed in organic growth experiments [49]. The chemical passivation of Si(111) produces a less hydrophilic surface with a contact angle (CA) of 81 ± 2 degree [101], close to hydrophobic limit (90 degree). On the other hand, small islands slightly separated or large islands largely separated are formed on the SiO_2_ surface if the deposition rate is relatively high or low, respectively.

As the growth proceeds, bricks of Voronoi tessellation including neighbouring islands begin to overlap. Monomers landing in the overlapping region have equal probability of being captured by either of the islands. This reduces the supply of monomers to each island, and limits further extension of fractal branches. A similar mechanism was proposed by Pratontep et al. [38] to explain the compact shape observed at relatively high deposition rates.

The 6T growth discussed in this work utilises a native SiO_x_ substrate that is more hydrophilic (CA = 58 ± 2 degree) than H-Si(111). Hence, as proved in reference [49], an increased island density is expected. The island density is also increased by the geometrical configuration of the in situ UHV system because of the OME cell has an angle of 60° with respect to the unit vector normal to the substrate plane. This inclination increases the effective deposition rate on the substrate by a factor ~10 in comparison to an OME cell *quasi* orthogonal to the substrate (12° in standard UHV flanges, see Appendix A). The combination of these two effects causes a higher island density, i.e., ~22 μm^−2^ at 25 °C and ~15 μm^−2^ at 50 °C, compared to the ones reported in reference [49] (~1.5 × 10^−3^ μm^−2^ on H-Si(111) and ~4.5 × 10^−2^ μm^−2^ SiO_2_). A higher island density provokes the overlap of neighbouring bricks thus promoting a compact or dendritic island shape.

Regardless *T_S_*, the trend of island shapes (*D_f_*) across MLs is homogeneous although a difference between native SiO_x_ and 6T layers is morphologically clear. As shown in Figure 4, 6T islands at the native SiO_x_ interface evolves from compact to dendritic shape for increasing deposited molecules. The literature on similar experiments performed with organic and inorganic materials is contradictory: some results agree with the shape evolution therein described [42,82,102,103], in others is observed the opposite, i.e., islands evolve from dendritic to compact shape [85,86,88,104].

Two theoretical works explain our observations considering both the presence of other islands deposited on a substrate with a moderate diffusion length [44] and relatively small value of the ratio *D*/*F* where *D* is the diffusion rate of monomers on the surface and *F* is the molecular flux (i.e., the deposition rate) [105]. In 6T experiments, the hydrophilic native SiO_x_ substrate reduces the diffusion length (i.e., *D*) whereas a high deposition rate increases the island density (presence of other islands) and reduces the ratio *D*/*F*. Accordingly, monomer incorporation is favoured with respect to edge diffusion along the molecular deposition, so the island shape evolves from compact to dendritic.

The evolution of island shape observed on native SiO_x_ is kept for all 6T MLs. In the next layers, *D_f_* increases slightly for comparable partial surface coverage *Θ_p_* (see Figure 4). As described above, the second ML grows following the FM growth mode. As well as the native SiO_x_ substrate, a complete 6T ML has a CA = (57 ± 1) degree [37,106], therefore the diffusion length can be assumed comparable with the one on native SiO_x_. Conversely, the island density is reduced to ~6 μm^−2^ at 25 °C and ~3 μm^−2^ at 50 °C. Being the islands more separated, a dendritic shape is favoured with respect to compact.

After the second ML, the OTF proceeds to the VW growth mode. The progressive dendritic shape in the third and fourth MLs agrees to the schematic of Luo et al. [47] where the ESB self-limits the underlayer islands promoting a more dendritic (even fractal) shape.

This shape evolution is similar for both *T_S_*_,_ although, at higher temperatures, islands are more dendritic (see Figure 4). This is clearly enhanced by the increase in surface diffusion, faster at 50 than at 25 °C, in agreement to Pimpinelli’s work, demonstrating that islands tend to be dendritic (or fractal) if, and only if, the edge diffusion is much slower than the surface one [107].

As shown in Figure 7, the differential entropy *S* evolves linearly with island shapes. This is not the first time where a linear correlation between Shannon entropy and fractal dimension was found. Chen and coworkers [108], for instance, studied city spatial distributions and, defining Shannon entropy through a box counting method, were able to find the same linear relationship. Following Figure 7, linear fits have a significant slope for the first ML whereas it is almost null for the next MLs. In the first ML, *D_f_* shifts from ~1.5 to ~1.66 (average values for 25 and 50 °C) with an entropy increase of 0.5 nats. For the next MLs, *D_f_* varies largely whereas *S* fluctuations are not significant (within ~0.1 nats for all MLs). As mentioned above, the *D_f_* change reflects an increase in the complexity of the island contour: a very simple contour would be a low *D_f_* that, conversely, increases for a very complex one. On the other hand, a change in *S* defines qualitatively the ‘roundness’ of finer details of the contour: e.g., for two *S* values, the lower one indicates a more circular contour, while the higher one corresponds to a more complex one, more difficult to be defined in a polar coordinate system (see Appendix A).

The cartoon of Figure 8 sketches the relationship between *S* and *D_f_* in Figure 7, where finer details of the contour and island shapes replace *S* and *D_f_*, respectively.

The differential entropy *S* in the *y*-axis shows an island with a circular contour at the inferior end that evolves into an island with comparable area but a more jagged contour at the superior end. On the other hand, the same island moves from compact to a dendritic/fractal shape in the *x*-axis. Such cartoon oversimplifies the differential entropy interpretation but suggests a geometrical interpretation of it: an increase in *S* is not only due to a polar symmetry loss moving from smooth to jagged contours. The roughness (fractality) of islands certainly increases *S* as proved by the entropy saturation obtained for higher *D_f_* (cp. Figure 4 and Figure 6).

Based on this scheme, the geometrical evolution of the island contour for increasing deposited molecules can be summarized as follows. At the early stage of the growth, compact islands form, with a smooth contour resulting in a relatively low value of both *D_f_* and *S*. As the coverage *Θ* increases, the island contour of the first ML slowly loses the polar symmetry while the contour roughness increases in reason of the monomer incorporation as inferred by the *D_f_* evolution. Progressively, islands reach a dendritic shape with *D_f_* close to the DLA value (dashed vertical line) while *S* reaches its relative maximum value (cp. to Figure 6). At this point, the dendritic shape of the islands dominates with respect to finer details of the contour highlighted by *S*. In the next MLs, the islands keep a dendritic shape (*D_f_* > 1.6) with *S* slightly varying around an average value of 5.1, since finer contour details are hindered by the dendritic/fractal shape of the island. This condition is preserved up to the fourth ML because compact islands, where the effect of finer details is pronounced, do not form anymore.

Lastly, this result suggests an imprint effect on the growth induced by the underneath substrate: the native SiO_x_ does not impose any significant shape constrain to the first ML growth, while a 6T monolayer promotes a dendritic shape constrain.

## 5. Conclusions

In summary, a new method of analysis, based on differential entropy, is herein introduced to describe and interpret the shape complexity of self-similar organic islands. The differential entropy, which can be described as an “angle entropy”, is a parameter rarely exploited in diverse scientific fields. However, it shows a high sensitivity to finer details of the island contour. In situ Atomic Force Microscopy allows imaging step-by-step the growth of 6T islands while deposition proceeds. The evolution of the island shape for increasing thickness can be investigated computing fractal dimension and differential entropy. On well-separated thin film islands, the fractal dimension shows a linear correlation versus the film thickness and a high sensitivity to changes in the island shape, while the differential entropy saturates at the completion of the first monolayer. Remarkably, a linear correlation between differential entropy and fractal dimension is observed. This enables one to discern the 6T growth on native SiO_x_ or 6T layer and helps to link growth phenomena to the shape complexity indicating, for instance, how many molecules are needed to recover information entropy. Further experiments to classify islands, like in computing science [51], can be envisaged in order to provide a deeper geometrical perspective to growth phenomena. On the other hand, this approach needs a theoretical support for a correct interpretation of such phenomena. Finally, this work might boost new research on the Ehrlich-Schwoebel barrier, whose dependence on the fractal dimension of islands has been already proven [6].

## Figures and Tables

**Figure 1 materials-14-06529-f001:**
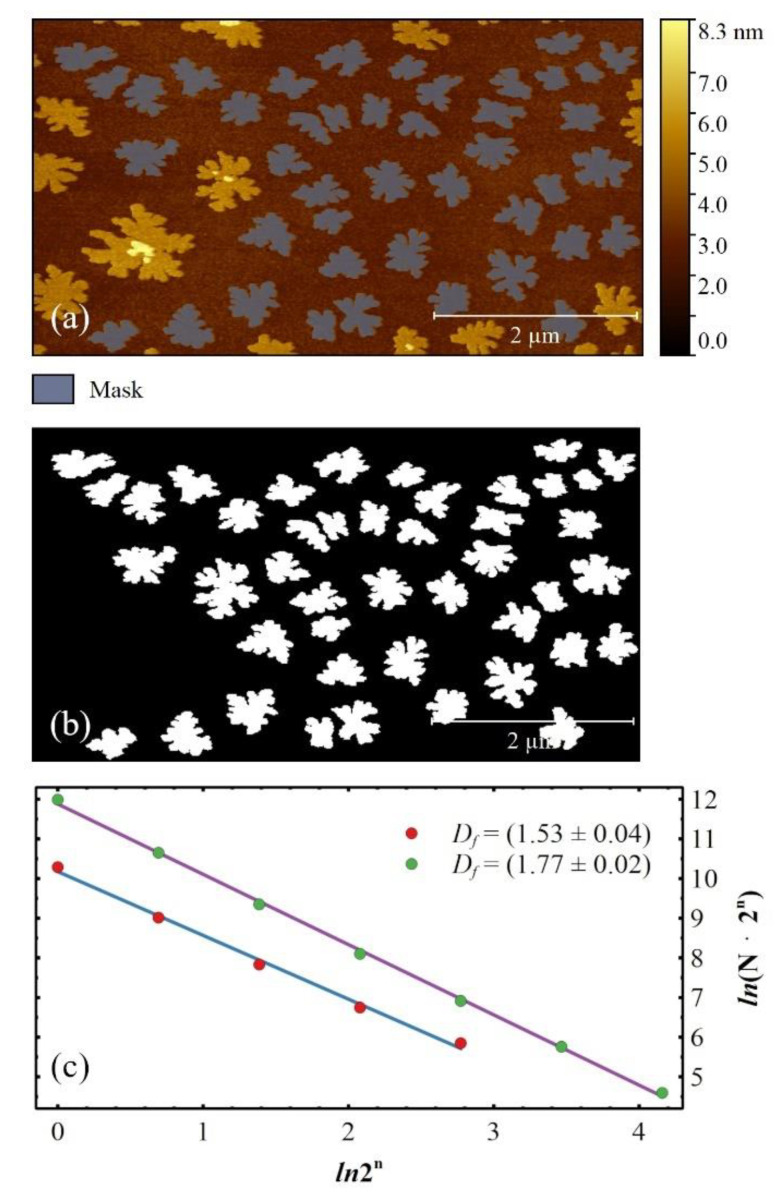
(**a**) Typical AFM image of 6T islands grown by Organic Molecular Beam Deposition (OMBD) on native SiO_x_ in UHV (*Θ* = 1.5 ML and *T_S_* = 50 °C). The bimodal growth is highlighted by both thresholding and filtering procedures (east bay coloured mask); (**b**) Binarized extracted mask with an average islands area of ~2850 px^2^, correspondent to a box size of ~55 px, closeness to 64 px, i.e., 2^6^ px as a power of 2; (**c**) Fractal dimension *D_f_* calculated for seven box sizes: 2^0^, 2^1^…2^6^ px (green dots). The natural logarithm (*ln*) of the number of covering boxes (*N*) of each size times the length of a box edge (2^n^) is plotted against the *ln*(2^n^). A straight-line with a negative slope *B* results, and *D_f_* is calculated as 1 − *B*. In the *x*-axis, the box size changes a few units, i.e., *ln*(2^6^) = 6·*ln*2~4.16, while in the *y*-axis, *N* spans 5 orders of magnitude, from few units for 2^6^ to ~1.6 × 10^5^ for 2^0^. The *D_f_* is affected by the average area of 6T islands, returning a smaller value of *D_f_* for a smaller average area (red dots, *Θ* = 1.08 ML and *T_S_* = 50 °C).

**Figure 2 materials-14-06529-f002:**
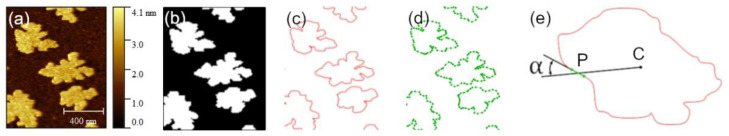
Graphical steps to obtain polygonal chains describing island contours. (**a**) Original AFM image; (**b**) two-levels bitmap image of well-identified islands are clearly; (**c**) Contours of the islands traced as a series of lines and Bezier curves; (**d**) Polygonal chains describing island contours (dots are vertexes of linear segments). For graphical purposes, linear segments of polygonal chains are 20 times lower than those used for calculating *S*. (**e**) Angle α between a segment centre of the island contour (green line with centre P) and the radial line to the centroid C of the island.

**Figure 3 materials-14-06529-f003:**
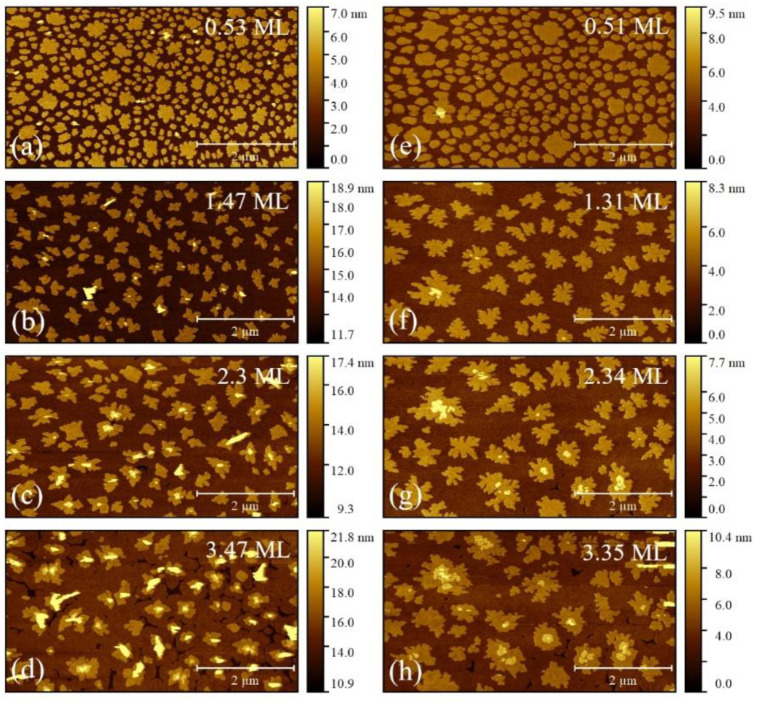
AFM images of 6T islands grown with OMBD in UHV on native silicon oxide at 25 °C (**a**–**d**) and at 50 °C (**e**–**h**). The surface coverage *Θ* for each AFM image is reported in the top-right corner. At each image, the same region was scanned, and the morphological evolution of OTFs can be followed along with the increasing coverage, up to the 4th ML. For graphical purposes, AFM images are cropped to 6.0 × 3.3 μm^2^, whereas some images (**b**–**d**) present a *z*-scale range starting from a height ≠ 0 nm because the underlying MLs are incomplete.

**Figure 4 materials-14-06529-f004:**
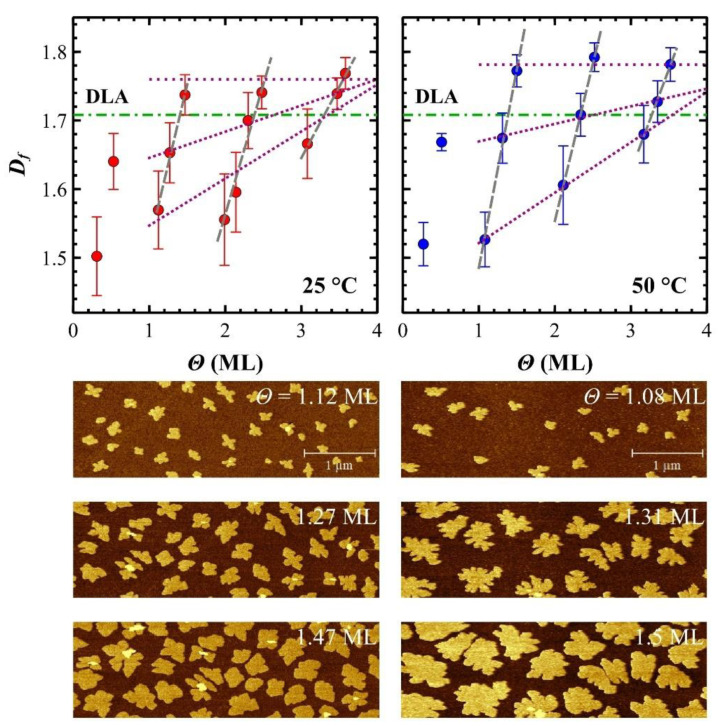
Fractal dimension plots for 25 °C (**left**) and 50 °C (**right**) referred to the DLA limit (dashed-dot lines). Illustrative examples of in situ AFM images for increasing partial coverage *Θp* below the aggregation regime (*Θp* < *n* + 0.5 where *n* = 0, 1, 2, and 3) for 25 °C (**left**) and 50 °C (**right**).

**Figure 5 materials-14-06529-f005:**
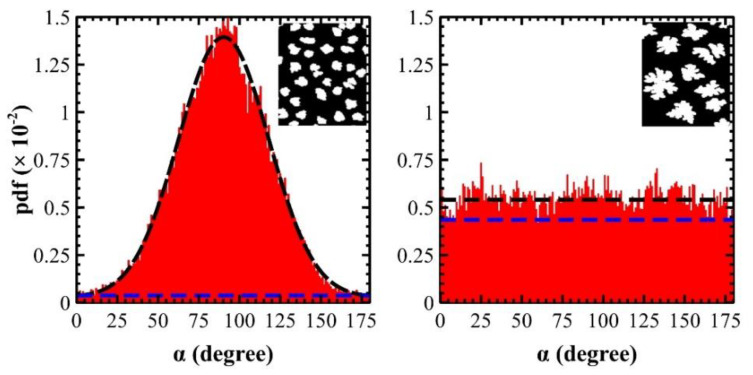
Illustrative pdf plots for compact (**left**) and dendritic (**right**) islands grown at 50 °C. The former is formed at the early stage of the growth (*Θ* = 0.28 ML), the latter at the first deposition step after the first ML (*Θ* = 1.29 ML). Both curves have a constant baseline of 0.00028 and 0.0045 (dashed blue lines) for compact and dendritic islands, respectively (see inset images). At 25 °C, Gaussian curve perfectly fits the pdf (dashed black lines) while the *quasi*-rectangular distribution shows a probability fluctuation around 0.0055.

**Figure 6 materials-14-06529-f006:**
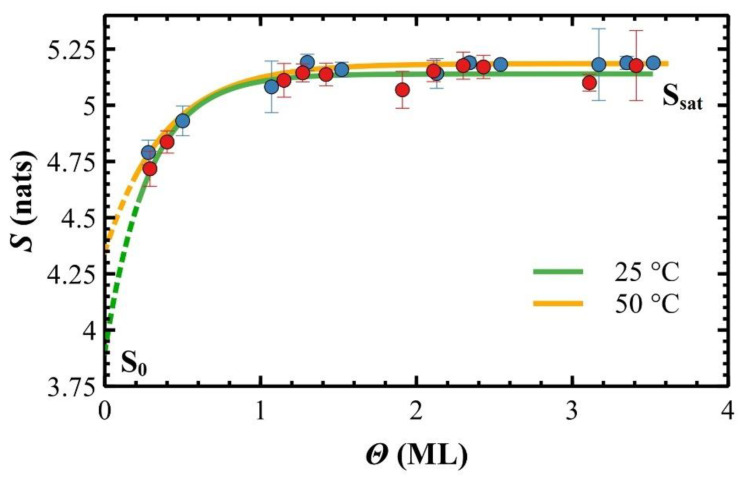
Data plots of *S* versus *Θ* at 25 and 50 °C fitted by an exponential *plateau* equation (Equation (2)).

**Figure 7 materials-14-06529-f007:**
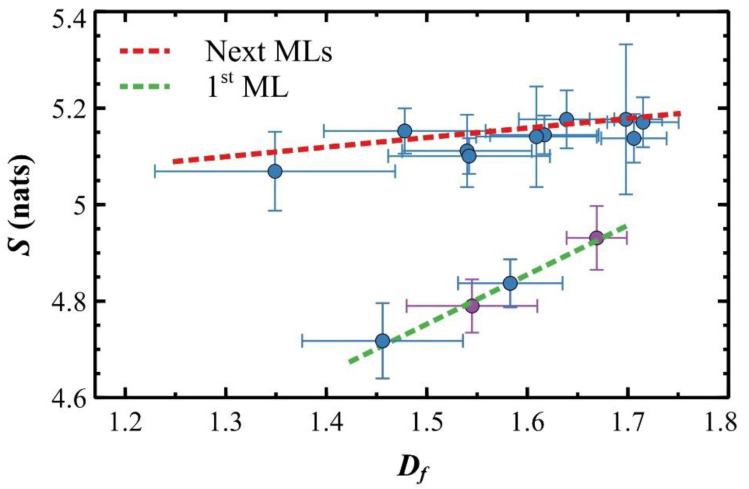
(Averaged) interfacial Shannon entropy *S* and fractal dimension *D_f_* plot with fitted linear correlations. All the data were obtained from the AFM images at 25 °C and 50 °C, averaging the values obtained from all the single islands.

**Figure 8 materials-14-06529-f008:**
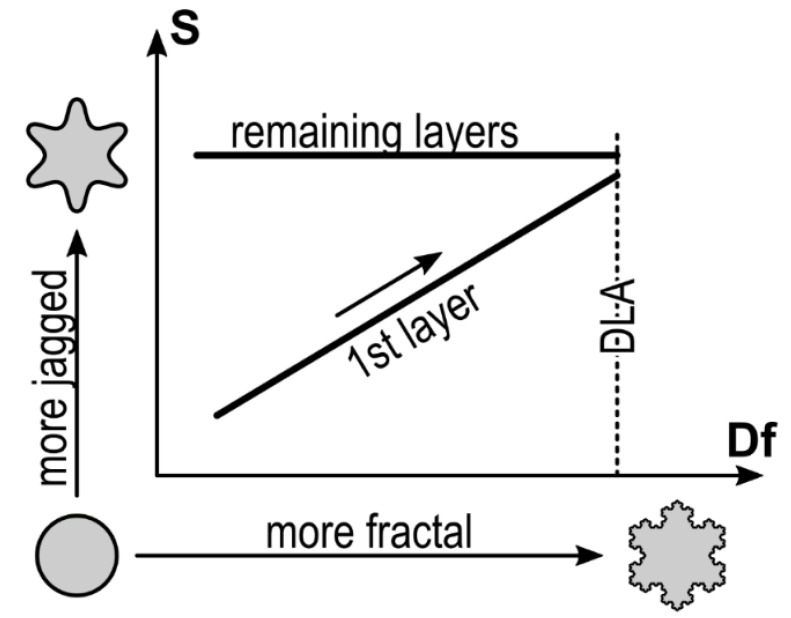
Qualitative interpretation of the geometric evolution of the island shape in terms of fractal dimension (*x*-axis) and differential entropy (*y*-axis) reported in Figure 7. The growth of the first (mono) layer is dictated by an initial formation of smooth circular islands, that progressively lose the circular symmetry and the contour becomes rougher (*S* increases due to a more jagged contour). The initial compact shape of the islands in the next MLs (remaining layers) is driven by the underneath 6T ML, that promotes a dendritic/fractal shape so that the initial circular symmetry is never recovered.

**Table 1 materials-14-06529-t001:** Data extracted from exponential *plateau* fitting by using Equation (2).

*T_S_* (°C)	*S_sat_* (Nats)	*S*_0_ (Nats)	*Θ_C_* (ML)
25	(5.14 ± 0.02)	~3.80	(0.28 ± 0.08)
50	(5.186 ± 0.004)	~4.35	(0.38 ± 0.07)

## Data Availability

Python code is available in the website https://github.com/Cristiano1974a/Python_code.git (accessed on 25 August 2021) and distributed with GNU General Public License v3.0. The data presented in this study are available on request from the corresponding author.

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
