# Peer review of "Differential Entropy: An Appropriate Analysis to Interpret the Shape Complexity of Self-Similar Organic Islands"

_materials, 2021, doi:10.3390/ma14216529_

Round 1

Reviewer 1 Report

The paper introduces a new parameter, differential entropy, to describe the shape complexity of the initial growing of organic thin film on SiOx substrate. Although it is a more phenomenal description, such new parameter is rationalized by correlating it with the often-used parameter, fractional dimension. The paper reveals the significant difference of the differential entropies between the first layer and the later layers in the organic film growing, which is deemed as the most innovative idea. In addition, the paper well summarizes the current works in theories on the thin film growth. Therefore, it will be a good reference for scientists in this field and hence worthy to be published.

Nevertheless, there are some format mistakes in the paper which make it difficult to follow in somewhere. Here are the lists that I could find:

  • In line 154, Figure 1, the abbreviation of OMBD was not specified.
  • In line 249, Figure 4, 3 fitting lines should correspond to 3 different thicknesses. They should be labeled. In addition, the data points should be different colors for different thicknesses at both temperatures. Also, the data for y-axis should be 1.5, 1.6, … etc, not 1,5 1,6  etc… and the same typos in Figure 5.
  • In line 276, Figure 5, “a constant baseline of 0.00028 and 0.0045 (dashed 276 blue lines)”. However, judged from both figures, the values are much larger.
  • In line 321, Table 1, the initial entropy, S0(nats) are ~ 3.80 and ~ 4.35 for two different temperatures. However, in line 335, these two values are 2.8 and 3.2 with errors, which are not presented in Table.
  • In line 337, the sentence “At 50 oC, for example, S0 shifts from ~ 0.7 nats for Θ ~ 0.25 ML (more compact islands) to ~ 4.2 nats for 338 Θ ~ 1.31 ML (more dendritic islands).” It is confusing: from the Figure 6, the S0 is the fitting parameter from different thicknesses at 50 oC and should be just one value. Why here S0 have different values at different Θ?
  • In line 460, “the lowest one” should be “the lower one”. Just English typo. Same in line 461, “the highest one” should be “the higher one”.
  • In line 471, “The cartoon of Figure 8 …”, but there is no Figure 8 in the draft.

In summary, this paper presents some novelty and a good summary in organic thin film growth.  It is recommended to publish after the author addresses the above questions.

Author Response

Reviewer_1 Report

The paper introduces a new parameter, differential entropy, to describe the shape complexity of the initial growing of organic thin film on SiOx substrate. Although it is a more phenomenal description, such new parameter is rationalized by correlating it with the often-used parameter, fractional dimension. The paper reveals the significant difference of the differential entropies between the first layer and the later layers in the organic film growing, which is deemed as the most innovative idea. In addition, the paper well summarizes the current works in theories on the thin film growth. Therefore, it will be a good reference for scientists in this field and hence worthy to be published.

Nevertheless, there are some format mistakes in the paper which make it difficult to follow in somewhere. In summary, this paper presents some novelty and a good summary in organic thin film growth.  It is recommended to publish after the author addresses the above questions.

Authors’ comment: We would like to thank Referee_1 to recognize our effort into correlate fractal dimension and differential entropy, although the latter induces a phenomenological interpretation of islands shapes. We have appreciated both comments and suggestions he/she raised and, following them, we return a more accurate manuscript. As reported below, questions and comments are answered constructively to the best of our understanding. Herein, changes in the manuscript text are reported in Italic font style in this document, while they are highlighted within the revised version of the manuscript text.

Reviewer_1

In line 154, Figure 1, the abbreviation of OMBD was not specified. 

Authors’ action: Abbreviation explained “Organic Molecular Beam Deposition (OMBD)”.

Reviewer_1

In line 249, Figure 4, 3 fitting lines should correspond to 3 different thicknesses. They should be labeled. In addition, the data points should be different colors for different thicknesses at both temperatures. Also, the data for y-axis should be 1.5, 1.6, … etc, not 1,5 1,6 etc… and the same typos in Figure 5.

Authors’ comments and replies: The referee should consider that AFM images are collected in situ and step-by-step, so the film thickness Θ (in monolayer, ML) is continuously measured. Accordingly, Θ reported in x-axis is obtained by a unique measurement along the whole film thickness (4 ML) with steps of 0.25 ML. The fitting lines Df vs Θ are therefore in function of the thickness, following a function Df = k·Θ + C where k is the line slope and C is a constant. As suggested by the referee, three different lines colors might highlight MLs differences but they risk to confuse readers about the continuous film growth. Two colors, red and blue, are already used for different temperatures (25 and 50 °C). We agree with the referee to the decimal notation, this is due to the software Veusz that takes automatically the Italian notation (in Italy, comma is used for decimal). All figures where comma is present, i.e. Fig. 4, 5, 6, and 7, are corrected.

Reviewer_1

In line 276, Figure 5, “a constant baseline of 0.00028 and 0.0045 (dashed 276 blue lines)”. However, judged from both figures, the values are much larger. 

Authors’ reply: The referee has probably forgot to multiply by 10-2 the measurable values of the constant baselines extracted from Figure 5, as indicated in y-axis.

Reviewer_1

In line 321, Table 1, the initial entropy, S0 (nats) are ~ 3.80 and ~ 4.35 for two different temperatures. However, in line 335, these two values are 2.8 and 3.2 with errors, which are not presented in Table. 

Authors’ comments and actions: As explained in the main text, S0 reported in Table 1 is graphically extracted by intercepting the exponential plateau function with the y-axis. This method is quite rough, for this reason S0 reported in Table 1 are approximated values. On the other hand, as explained in the main text, “The initial entropy S0 depends on pdfs constant baselines, in average (2.8 ± 1.1) nats for 25 °C and (3.2 ± 1.3) nats for 50 °C, intervals including S0 graphically extrapolated from fitting curves.”, i.e. S0 statistically calculated from pdfs baselines confirms S0 graphically extrapolated from fitting curves. Anyway, we recognize that this paragraph is quite confused, so it is reshaped in the following way (from line 335): “The large difference of S0 for 25 and 50 °C (see Table 1) corresponds to compact or dendritic shapes, respectively (at the scan-size analyzed in these experiments). Graphical extrapolation of S0 from fitting curves is clearly weak since is basically supported by only two datapoint for each TS (only two thickness Θ satisfy the experimental condition of separated islands at Θ < 1 ML) and because the fitting function has the highest derivative at Θ = 0. Tentatively, the initial entropy S0 could be dependent on pdfs constant baselines. By averaging the constant baselines of pdfs sets (11 pdfs for each TS), S0 is (2.8 ± 1.1) nats for 25 °C and (3.2 ± 1.3) nats for 50 °C. These intervals include S0 graphically extrapolated from fitting curves but, although more statistically relevant, this result is not supported by a theoretical explanation.”,  

Reviewer_1

In line 337, the sentence “At 50 °C, for example, S0 shifts from ~ 0.7 nats for Θ ~ 0.25 ML (more compact islands) to ~ 4.2 nats for 338 Θ ~ 1.31 ML (more dendritic islands).” It is confusing: from the Figure 6, the S0 is the fitting parameter from different thicknesses at 50 °C and should be just one value. Why here S0 have different values at different Θ?

Authors’ comment and actions: As reported above, we agree to the referee that the paragraph including the cited sentence is quite confused. The paragraph is reshaped in the following way (from line 335): “The large difference of S0 for 25 and 50 °C (see Table 1) corresponds to compact or dendritic shapes, respectively (at the scan-size analyzed in these experiments). Graphical extrapolation of S0 from fitting curves is clearly weak since is basically supported by only two datapoint for each TS (only two thickness Θ satisfy the experimental condition of separated islands at Θ < 1 ML) and because the fitting function has the highest derivative at Θ = 0. Tentatively, the initial entropy S0 could be dependent on pdfs constant baselines. By averaging the constant baselines of pdfs sets (11 pdfs for each TS), S0 is (2.8 ± 1.1) nats for 25 °C and (3.2 ± 1.3) nats for 50 °C. These intervals include S0 graphically extrapolated from fitting curves but, although more statistically relevant, this result is not supported by a theoretical explanation.

Reviewer_1

In line 460, “the lowest one” should be “the lower one”. Just English typo. Same in line 461, “the highest one” should be “the higher one”.

Authors’ actions: Corrected.

Reviewer_1

In line 471, “The cartoon of Figure 8 …”, but there is no Figure 8 in the draft.

Authors’ comments and actions: Thanks to the referee comment, we found two errors in the caption of Figure 8, which was erroneously indicated as Figure 7. Moreover, a reference to Figure 6 in the caption text is corrected to Figure 7. On the other hand, Figures references in the main text are correct.

Reviewer 2 Report

I can recommend the publication of the manuscript after the following minor comments:
This manuscript must be verified by a native English speaker. There are some typos and minor grammatical errors.
Page 3, line 100: Why it was chosen as unit MLs? Kindly insert corresponding explanations for this abbreviation.
Page 3, line 134: Why it was chosen the box-counting method? Give more details about advantages and disadvantages, as well the limits of this method.
Page 9, line 309: How was chosen “exponential plateau function”? Could you insert more details?
Page 10, lines 374-379: insert corresponding references.
If possible, insert some parameters about the surface morphology of samples.
References are not written according the Guide of Authors (ref. [17], [20], [79], [95], [98], [100], [105], [108], and so on).
6. Authors may consider citing the following reference:
[1] Ş. Ţălu, Micro and nanoscale characterization of three-dimensional surfaces. Basics and applications. Napoca Star Publishing House, Cluj-Napoca, Romania, 2015.

Author Response

Reviewer_2 Report

I can recommend the publication of the manuscript after the following minor comments:
This manuscript must be verified by a native English speaker. There are some typos and minor grammatical errors.

Authors’ comment and actions: We would like to thank Referee_2 for recommending our work for publication in “Materials”. Franco Dinelli, an Italian researcher with a PhD in Oxford and several years of postdoc in UK and USA, has polished the manuscript English. As required by the referee, this proofreading has improved the manuscript clarity , especially in the Introduction and Conclusions sections. Moreover, the title has been slightly changed into: “Differential entropy: an appropriate analysis to interpret the shape complexity of self-similar organic islands”; in our opinion, this title is more focused on the topic discussed in the manuscript. 

Reviewer_2

Page 3, line 100: Why it was chosen as unit MLs? Kindly insert corresponding explanations for this abbreviation.

Authors’ comments and actions: The monolayer (ML) unit for measuring the film thickness is commonly used in organic films grown by subliming small organic molecules (see the first 30 references in the manuscript), but we agree to referee because it is not properly defined. In addition to the description of Section 2.1 of the “Materials and methods”: “The OTF thickness was measured in MLs1, so each layer composing the OTF can be expressed in terms of an equivalent surface coverage Θ [15].”, a footnote is added: “The OTF thickness is measured in monolayer units (ML), by defining one monolayer as a layer of ordered and packed molecules (almost) orthogonal to the SiOx surface that completely covers it (Θ = 1). In the case of 6T, each monolayer correspond to a thickness of 2.4 nm [22].”.  

Reviewer_2

Page 3, line 134: Why it was chosen the box-counting method? Give more details about advantages and disadvantages, as well the limits of this method.

Authors’ comments: The box-counting method was chosen rather than the islands-perimeter one, a method often used in this kind of molecular system (see Ref. Valle, F.; Brucale, M.; Chiodini, S.; Bystrenova, E.; Albonetti, C. Nanoscale morphological analysis of soft matter aggregates with fractal dimension ranging from 1 to 3. Micron 2017, doi:10.1016/j.micron.2017.04.013) because of the maximum lateral scan-size permitted by our AFM,  as explained in the main text from line 124 to line 164, together with the cited references.

Reviewer_2

Page 9, line 309: How was chosen “exponential plateau function”? Could you insert more details?

Authors’ comments: As claimed in the manuscript (lines 310-311) “Data are empirically fitted with the exponential plateau function”, the function is therefore empirical because there is not a theoretical basis for sustaining this choice. To our best knowledge, our manuscript is the first example in the literature showing a correlation between fractal dimension and differential (information) entropy of organic islands therefore no theoretical descriptions for explaining this correlation were published to date.

Reviewer_2

Page 10, lines 374-379: insert corresponding references.
If possible, insert some parameters about the surface morphology of samples.
References are not written according the Guide of Authors (ref. [17], [20], [79], [95], [98], [100], [105], [108], and so on).

Authors’ comments: References are inserted as required while, in our opinion, other morphological parameters like, for instance, roughness are useless for this investigation where a geometrical approach for study the shape complexity was chosen (as stated in the Introduction). About the bibliography, it is managed automatically by Mendeley (Download Mendeley Reference Manager For Desktop | Mendeley) with the “Materials” citation style. By comparing our manuscript to other papers published in Materials, apparently we don’t found formatting errors. Anyway, post-reviews process will correct possible typographic errors.

Reviewer_2

Authors may consider citing the following reference:
[1] Ş. Ţălu, Micro and nanoscale characterization of three-dimensional surfaces. Basics and applications. Napoca Star Publishing House, Cluj-Napoca, Romania, 2015.

Authors’ actions: Reference inserted.
